# Effect of 2% Chlorhexidine Following Acid Etching on Microtensile Bond Strength of Resin Restorations: A Meta-Analysis

**DOI:** 10.3390/medicina55120769

**Published:** 2019-12-02

**Authors:** Tasnim Hamdan-Nassar, Carlos Bellot-Arcís, Vanessa Paredes-Gallardo, Verónica García-Sanz, Agustín Pascual-Moscardó, José Manuel Almerich-Silla, José María Montiel-Company

**Affiliations:** Stomatology Department, University of Valencia, 46010 Valencia, Spain; tasham@alumni.uv.es (T.H.-N.); carlos.bellot@uv.es (C.B.-A.); vanessa.paredes@uv.es (V.P.-G.); agustin.pascual@uv.es (A.P.-M.); jose.m.almerich@uv.es (J.M.A.-S.); jose.maria.montiel@uv.es (J.M.M.-C.)

**Keywords:** chlorhexidine, microtensile bond strength, adhesion, dental adhesive, acid etching

## Abstract

*Background and Objectives:* The aim of this systematic review was to examine the effect of 2% chlorhexidine following acid etching on the microtensile bond strength of resin restorations for different follow-up times. *Materials and Methods:* A thorough search of PubMed, Scopus, and Embase databases were conducted. In vitro experimental studies or in vivo studies published up to December 2018 with an experimental group treated with a 2% chlorhexidine solution following acid etching and a control group were included, wherein the final restoration used a resin composite in both the groups. *Results:* Twenty-one articles were identified for qualitative analysis and 18 for meta-analysis. The difference in the means of microtensile bond strength between the two groups was calculated for the different follow-up times. The differences were significant for 6 months (4.30 MPa; 95% CI 2.72–5.89), 12 months (8.41 MPa; 95% CI 4.93–11.88), and 2–5 years including aged and thermocycling samples (9.08 MPa; 95% CI 5.36–12.81). There were no significant differences for the type of adhesive used. A meta-regression model showed a significant effect of time on the microtensile bond strength. *Conclusions:* The application of a 2% chlorhexidine solution after acid etching increased the microtensile bond strength significantly for follow-up times of 6 months or more. The adhesive type had no influence.

## 1. Introduction

Chlorhexidine (CHX) is now recognized as the main chemical agent for controlling and preventing biofilm and inhibiting gingivitis [1,2]. Chlorhexidine is a cationic bisbiguanide with a mode of action based on breaking down the cytoplasmic membrane of microorganisms by disturbing their osmotic balance, causing precipitation of the cell contents. It is a wide-spectrum antibacterial agent which is bacteriostatic at low concentrations and bactericidal at high concentrations [1,2]. The risk of resistance of oral bacteria to chlorhexidine and potential mechanisms conferring this resistance or even cross-resistance to the antibiotic is little known. Moreover, there is also little awareness about the risk of chlorhexidine resistance among the dental community even though chlorhexidine has been widely used in dental practice as the gold-standard antiseptic [3,4].

The role of chlorhexidine as a synthetic protease inhibitor was evaluated in the study by Carrilho et al., 2007 [5]. The ability of chlorhexidine to dose-dependently inhibit the collagenolytic activity of metalloproteinases 2 and 8 and cysteine cathepsins present in the human dentin–pulp complex or in inflammatory diseases, such as periodontitis, has been described [5,6,7].

Metalloproteinases (MMPs) are present within the human dentin matrix and are involved in dentinogenesis and in caries progression mechanisms. Human dentin matrices exhibit collagenolytic and gelatinolytic activity when prepared with acid and bonding agents. This event is responsible for the disruption of collagen fibrils. The incomplete infiltration of these fibrils into the hybrid layers of the resin–dentin bond leads to bond break down over time [8].

Four types of MMPs have been identified in dentin: MMP-2, MMP-9 (gelatinases), MMP-8 (collagenases), and MMP-20 (enamelysin) [4]. MMP-2 predominates over the rest [9]. MMPs in the dentin are activated by acids produced by cariogenic bacteria, causing destruction of the collagen matrix in carious processes. In addition, acid etching prior to the placement of a restoration is also capable of breaking the collagen matrix by activating the MMPs in the dentin that are slowly released over time [4,10,11].

Chlorhexidine inhibits the action of MMP-2 and MMP-9 by a chelating mechanism, whereas in the case of MMP-8, chlorhexidine may interact with the essential sulfhydryl and/or cysteine groups present in the active site of MMPs [8].

The longevity of dental restorations depends on the bond between the restoration and the tooth. This union is a hybrid layer formed by a mixture of collagen, resin, residual water, and hydroxyapatite crystallites generated when the resin penetrates into the intertubular dentin and the dentin tubules [12]. The formation of this hybridized layer is not enough to guarantee the stability of the layer over time [12,13].

In order to maintain the stability of the hybrid layer, chlorhexidine can be applied. Chlorhexidine has a high substantivity due to the release of positively charged molecules on the treated surfaces. The demineralized fibrils of dentinal collagen can be united in a very short period of time (15–60 s), this being enough to guarantee long-term bonding [14,15].

No side effects have been described for chlorhexidine (such as brown staining or unpleasant taste alteration) in short-term applications. Therefore, chlorhexidine would not have any negative effect when used in the adhesion process [16].

Carrilho observed that application of an aqueous CHX solution after acid-etching resulted in stable resin–dentin bonds after approximately 14 months [5]. Some clinicians apply 2% CHX for 60 s to acid-etched dentin during resin bonding in an attempt to increase the durability of resin–dentin bonds by inhibiting endogenous MMPs in the dentin matrix. This method is easy to adopt, and will likely be the first to gain wider acceptance.

Bond strength testing of adhesive systems is considered a reliable predictor of the longevity of dental restorations. Bond strength can be measured by different types of testing methods. Microtensile bond tests provide many advantages over the shear tests, such as better control of regional differences, better economic use of teeth, and better stress distribution at the true interface. The drawbacks include difficulty in measuring bond strengths lower than 5 MPa, difficulty in fabricating specimens with consistent geometry, and easy damage of specimens [17].

The aim of the study was to analyze the available evidence on the effect of 2% chlorhexidine after acid etching on the microtensile bond strength resistance of resin restorations over time: immediately, at 24 h, at 6 months, at 12 months, between 2–5 years, and in aged or thermally cycled restorations.

The research (PICO) question was: do resin restorations pretreated with 2% chlorhexidine after etching have greater long-term microtensile bond strength stability compared to their untreated counterparts?

## 2. Experimental Section

### 2.1. Study Design

This study consisted of systematically reviewing published studies that addressed the effect of chlorhexidine on the microtensile bond strength in resin restorations.

### 2.2. Inclusion and Exclusion Criteria

The criteria for inclusion accepted all articles that met the following requirements:Published up to 31/12/2018 in PubMed, Scopus, and Embase.In vitro experimental studies or in vivo studies.Control group and experimental group, the latter treated with 2% chlorhexidine following acid etching.English language.

The exclusion criteria were:Studies that did not use a resin composite for the final restoration.Self-etch adhesive system.Caries-affected dentin.Eroded dentin.

### 2.3. Data Sources and Search Strategy

Searches were made in the electronic databases to identify findings published up to December 2018.

The search terms were “dental bonding”, “dentin bond”, “dentin bonding”, “dentin bond strength”, “dental bonding stability”, “dentin bond strength stability”, “dentin microtensile bond strength”, “dentin bond strength durability”, “dentine bonding”, “chlorhexidine”, “chlorhexidine gluconate” and “chlorhexidine digluconate”.

In order to identify other potentially valid studies, manual searches were made in journals directly related to the research objective, namely, Operative Dentistry, Dental Materials, Journal of Dentistry, Journal of Dental Research, Journal of Adhesion Science and Technology, International Journal of Adhesion and Adhesives, International Journal of Contemporary Dentistry, The Journal of Contemporary Dental Practice, The Journal of Adhesive Dentistry, Dental Materials Journal, and Restorative Dentistry.

The search strategy used in MEDLINE (accessed through PubMed, using MeSH terms), Embase (using Emtree terms), and Scopus combined OR and AND commands to obtain the greatest possible quantity of information.

For MEDLINE: (((“dental bonding”(eSH Terms) OR (“dental”(All Fields) AND “bonding”(All Fields)) OR “dental bonding”(All Fields)) OR ((“dentin”(MeSH Terms) OR “dentin”(All Fields)) AND bond(All Fields)) OR ((“dentin”(MeSH Terms) OR “dentin”(All Fields)) AND (“object attachment”(MeSH Terms) OR (“object”(All Fields) AND “attachment”(All Fields)) OR “object attachment”(All Fields) OR “bonding”(All Fields))) OR ((“dental bonding”(MeSH Terms) OR (“dental”(All Fields) AND “bonding”(All Fields)) OR “dental bonding”(All Fields)) AND stability(All Fields)) OR ((“dentin”(MeSH Terms) OR “dentin”(All Fields)) AND bond(All Fields) AND strength(All Fields) AND stability(All Fields)) OR (microtensile(All Fields) AND bond(All Fields) AND strength(All Fields))) AND ((“chlorhexidine”(MeSH Terms) OR “chlorhexidine”(All Fields)) OR (“chlorhexidine gluconate”(Supplementary Concept) OR “chlorhexidine gluconate”(All Fields)) OR (“chlorhexidine gluconate”(Supplementary Concept) OR “chlorhexidine gluconate”(All Fields) OR “chlorhexidine digluconate”(All Fields)))) AND (“0001/01/01”(PDAT) : “2018/12/31”(PDAT)).

For Embase: (dentin AND bond AND strength OR ‘dentin bonding agent’ OR (dentin AND bond AND strength AND stability)) AND chlorhexidine AND (1966–2018)/py.

For Scopus: dental AND bonding AND durability AND adhesive AND microtensile AND bond AND strength AND in AND vitro AND resin AND composite AND dentin AND chlorhexidine AND (EXCLUDE (PUBYEAR, 2019)).

### 2.4. Quality Assessment

The quality of the articles included in the review was assessed on the Jadad scale, which is used to evaluate the methodological quality of experimental studies. The scale is composed of 5 items which assess the biases related to randomization, double-blinding, withdrawals and dropouts, appropriate randomization, and appropriate blinding. The score can range from 0–5 points. A score of 5 points is considered to be of high quality. An article that scores less than 3 points is considered to be of poor quality.

### 2.5. Data Mining

Two reviewers independently assessed the articles. In the event of disagreement, they reached a consensus, but if they continued to disagree they consulted a third reviewer.

The following data items were recorded for each article:Adhesive type and brand.Composite type and brand.Sample size (*n*).Follow-up time for each sample group.Mean microtensile bond strength of each sample group measured in megapascals (MPa), with standard deviation (SD). Microtensile bond strength was the main variable in the present study.

### 2.6. Statistical Analysis of the Data

The data were studied through a meta-analysis performed with the Comprehensive Meta-Analysis v3.0 software (Biostat, Inc. Englewood, CO, USA). The studies were combined using random effects models. The Q-test was used to assess the heterogeneity of the studies included in the meta-analysis. When *p* < 0.1, the meta-analysis was considered heterogeneous. Heterogeneity was also measured using I^2^. An I^2^ score of 25–50% indicated low heterogeneity, over 50% and up to 75% moderate heterogeneity, and over 75% high heterogeneity.

The effect size was estimated by calculating the difference in the means of microtensile bond strength between the experimental group and the control group. The meta-analysis was considered significant when the *p*-value of Z-test was below 0.05. The meta-analysis was represented by forest plots. In addition, the study was completed with a meta-regression analysis with the maximum likelihood model and a scatter plot was obtained.

### 2.7. Publication Bias Assessment

The publication bias was assessed by a funnel plot. The x-axis reflects the effect of the treatment, in this case the differences in means, and the y-axis represents the degree of confidence in the results of the studies, in this case represented by the standard error. Under the random effects model and using the "trim-and-fill" method, which is able to impute new studies to obtain a symmetry in the funnel plot, the difference in means was estimated and compared with that obtained previously in the overall meta-analysis.

The publication bias was also assessed through the classic fail-safe number and Egger’s regression intercept. The classic fail-safe number is interpreted as the number of non-significant studies that would need to be included in the meta-analysis in order for the significant *p*-value resulting from the calculation to cease to be significant. In the absence of publication bias, the Egger’s regression intercept has a 95% CI that includes 0 and a *p*-value > 0.

## 3. Results

The search identified 449 articles in MEDLINE, 289 in Embase, 439 in Scopus, and 3 through manual search. Duplicate references were excluded. After screening by title and abstract and reading the full text, 21 were included in the qualitative synthesis and finally 18 in the quantitative synthesis (meta-analysis) because the articles did not present data available or could not be combined because the standard deviation was missing (Figure 1). Table 1 presents the data extracted from the selected articles. All the studies, when evaluated by the Jadad quality scale, presented medium-low scores (Table 2).

### 3.1. Overall Meta-Analysis of Follow-Up Times

The overall meta-analysis included 18 studies with 44 rows, because there were various subgroups in the same study. This meta-analysis studied the microtensile bond strength, irrespective of follow-up time (Figure 2). The bond strength of the experimental group—the one treated with chlorhexidine—was 3.49 MPa (95% CI 2.26–4.72) greater than that of the control group. The meta-analysis did show significance (Z-value = 5.58; *p* = 0.000). The Q-test value was 468.2, with *p* = 0.000, and the I^2^ score was 90.8%, so it was considered highly heterogeneous. To analyze the influence of time on the microtensile bond strength, the studies were grouped based on the follow-up times.

### 3.2. Meta-Analysis by Follow-Up Time: Immediate

The second meta-analysis was performed with the subgroup where the microtensile bond strength had been measured immediately (Figure 3). It included 12 studies. The Q-test value of 131.31 (*p* = 0.000) and the I^2^ score of 74.8% indicated moderate heterogeneity. The difference in means was 0.60 MPa (95% CI −1.53 to 2.72). The data were not significant (Z-value = 0.55; *p* = 0.583).

### 3.3. Meta-Analysis by Follow-Up Time: 24 h

The third meta-analysis covered the 6 studies that analyzed 24 h after placement (Figure 4). The Q-test value of 19.1 (*p* = 0.002) and the I^2^ score of 73.8% showed moderate heterogeneity. The difference between means was –1.38 MPa (95% CI −5.95 to 3.20). The meta-analysis did not show statistical significance (Z-value = −0.59; *p* = 0.555).

### 3.4. Meta-Analysis by Follow-Up Time: 6 Months

The 6-month subgroup included 9 studies (Figure 5). The Q-test value was 57.9 (*p* = 0.000) and the I^2^ score was 84.4%. The 4.30 MPa (95% CI 2.72–5.89) difference between the means was favorable to the chlorhexidine group and statistically significant (Z-value = 5.32; *p* = 0.000).

### 3.5. Meta-Analysis by Follow-Up Time: 12 Months

In the 12-month subgroup, 4 studies were included (Figure 6). The Q-test value was 9.31 (*p* = 0.025) and the I^2^ score was 67.8 % showing moderate heterogeneity. The difference in means was 8.41 MPa (95% CI 4.93–11.88) and presented statistical significance (Z-value = 4.74; *p* = 0.000).

### 3.6. Meta-Analysis by Follow-Up Time: From 2–5 Years, Aged, or Thermocycling

This meta-analysis included 7 studies (Figure 7). The aged subgroup was unified with the thermocycling subgroup, because in spite of the different nomenclatures chosen by each author in their article, both corresponded to the same process. This consisted of thermocycling accompanied by an increase in temperature, thus producing an aged specimen to be studied. The Q-test value was 50.7 (*p* = 0.000) and the I^2^ score was 88.1%. Like all the previous meta-analyses, these data presented heterogeneity. In this model, the difference in means was 9.08 MPa (95% CI 5.36–12.81) and presented statistical significance (Z-value = 4.78; *p*-value = 0.000). So, it may be stated that from 6 months onward, the microtensile bond strength of the chlorhexidine group was greater than that of the control group.

### 3.7. Meta-Regression

By using a regression model (random effects and maximum likelihood) with the difference of means as the dependent variable and the follow-up time and type of adhesive as covariates, a significant model was obtained (Q-value = 46.64, df = 13, *p* < 0.001, R^2^ = 0.61; Table 3). The value of the intercept was 3.73 (−3.89 to –11.41). Taking the mean difference in the immediate category as the reference category, the beta coefficients of the 6-month category (4.39 with 95% CI 1.20–7.58), 12-month category (6.66 with 95% CI 1.64–11.68), and 2–5 years and aged or thermocycling category (8.50 with 95% CI 4.89–12.11) presented high significance in the model, indicating an increase of microtensile bond strength after 6 months in the chlorhexidine group (Figure 8). When analyzed according to the type of adhesive used and taking SB (Single Bond) as the reference category, none of the adhesives showed significance in the model (Figure 9).

### 3.8. Publication Bias Assessment

The funnel plot (Figure 10) was quite symmetrical, showing studies with a high standard error and others with a low standard error. When the trim-and-fill method was used under the random effects model, the point estimated (black rhombus in Figure 10) remained unchanged with respect to the estimated point in the overall meta-analysis. The classic fail-safe number calculated 2560 as the number of studies that would be needed for the *p*-value to cease to be significant. The Egger’s recession intercept was 0.291, 95% CI −1.08 to –1.66 with a *p*-value of 0.436. For all these studies, the presence of a publication bias was not detected.

## 4. Discussion

Several studies support the use of chlorhexidine to inhibit dentinal MMPs as well as decelerating the loss of resin–dentin bonds [14,20,21]. Based on this hypothesis, the present review examined the function of 2% chlorhexidine on the inhibition of MMPs.

Most of the published articles analyzed different concentrations of chlorhexidine or other disinfectants and additionally, some combined different adhesive systems or teeth that presented caries-affected dentin or eroded dentin.

The collected data belonged to those groups in which chlorhexidine was applied after acid etching. This is justified since it has been observed that its application after etching increases the wetting capacity of the first dentin, thus improving adhesion [36]. Its application before etching is not effective because the bonding of chlorhexidine to mineralized dentin (and without engraving) is almost 80% lower than to demineralized dentin [37,38]. After acid etching, the dentinal tubules become larger when the peritubular dentin is lost, increasing its water content and facilitating the diffusion of the chlorhexidine into the dentin matrix [39]. Also, if chlorhexidine is applied before etching, most of it will be lost after acid washing and drying [16].

In the present research, the authors decided to study the adhesives that required prior etching with orthophosphoric acid, avoiding the self-etch adhesives since they have not been studied in depth and the results obtained are contradictory [40].

Several studies have shown that adhesives that require prior etching and washing with water (such as those included in this meta-analysis) achieve higher adhesive bond values than single-stage self-etching adhesives [41,42,43]. The reason is that weak acids possess the potential to activate the MMPs, particularly when their pH lies between 2.3 and 5, which is the case of many self-etch adhesives, making them very effective at activating gelatinase action [32]. Adhesion depends on the adhesive system used, with the non-simplified adhesive systems (etch and rinse) being more stable and effective than the simplified adhesive systems (self-etch) [44,45,46].

The long-term adhesion strength increases when the dentin is washed with chlorhexidine after etching with the 3-step etch-and-rinse adhesives [40,47,48].

Most of the included studies in this research were in vitro studies. Only one in vivo study [32] could be included due to the difficulty in carrying out these type of studies and due to ethical restrictions.

Chlorhexidine at 2% is one of the most investigated concentrations and it shows the greatest inhibitor effect on MMPs and increase in microtensile bond strength [45].

At concentrations of 0.05% or 0.2%, there is a certain degree of inhibitory effect of MMPs [6,36,48,49]; however, a clear relationship between the concentration of chlorhexidine and the increase in the bonding strength of dentin–resin bonds has not been demonstrated [44].

The studies included in the quantitative analysis showed greater mean microtensile bond strengths in the groups that employed chlorhexidine, which increased over longer follow-up times; however, not all the studies reported statistically significant differences.

The adhesion strength between the restoration and the dentin after the use of chlorhexidine was measured in MPa by the microtensile bond strength variable. This variable is calculated using the microtensile test in which a series of sticks/beams of tooth with a bond area cross-section of approximately 1 mm^2^ are prepared, tested, and examined under a stereomicroscope. The sticks were bonded with cyanoacrylate adhesive and subjected to traction with a cross-speed of 1 mm/min until failure occurred.

When chlorhexidine was used as an MMP inhibitor and its effect was measured immediately and over the long term, it promoted different effects on the microtensile bond strength of the resin restorations. This fact was observed in the data obtained from the meta-analysis which showed differences in microtensile bond strengths between the groups assessed immediately and the groups assessed at 24 h compared to the groups assessed at 6 months, 12 months, and 2–5 years including aged or thermally cycled samples.

The immediate follow-up subgroup meta-analysis showed no increase in the resin–tooth bond strength in the experimental group using 2% chlorhexidine compared to the control group. Indeed, the results even favoured the control group over the experimental group, although the data were not statistically significant. These findings are in line with the observations of several in vitro studies which occasioned the debate as to whether chlorhexidine might actually reduce the immediate bond strength [15,19,20,21,23,32,50]. Chlorhexidine application did not influence the immediate bond strength of conventional adhesives [21,51] and did not affect the adhesives used in the studies included in the present meta-analysis; however, 2% chlorhexidine was found to lead to a reduction in the immediate dentinal bond values of self-etching adhesives, such as Clearfil SE Bond and Clearfil S3 Bond [21]. In primary teeth, no beneficial effect of chlorhexidine has been observed [23].

In the 24 h subgroup meta-analysis, as in the immediate follow-up group, chlorhexidine application did not influence microtensile bond strength. This is in agreement with the findings of several authors [24,25,26,27,28,30,31].

One possible explanation is that a short period of time may be insufficient to detect the effects of the hydrolytic breakdown of the adhesive interface that leads to failure of the hermetic seal produced by the adhesive. Additionally, the dimensions of the dentinal surface to be broken down must be taken into account [24].

In the meta-analysis of the 6-month subgroup, a statistical significance was observed indicating an increase of 5.02 MPa in the microtensile bonding strength in the chlorhexidine group compared to the control group. This increase in adhesion strength after 6 months was also corroborated by a recent study in 2017 despite the fact that chlorhexidine concentration was lower than 2% [49].

In the 12-month subgroup, the present meta-analysis found statistically significant changes in microtensile strengths. The bond strength of the group treated with chlorhexidine was found to be 6.2 MPa greater than the control group.

The meta-analysis of the subgroup for 2 and 5 years also yielded significant results. As observed with the two previous subgroups, chlorhexidine favoured the microtensile bonding strength of the restoration (10.52 MPa higher than the control group).

This finding agrees with the results of various studies that reported increased bond strengths in the long-term following the use of chlorhexidine [20,31,52,53]. Breschi concluded that MMPs are responsible for the time-dependent breakdown of hybrid layers and that MMP inhibition performs an important role in improving the durability of resin–dentin bonds [20].

The last subgroup analyzed according to the follow-up time included the analysis of aged specimens and those subjected to thermocycling. Despite the different nomenclatures granted by each author, both conduct the same process. This process consists of temperature changes (5–55 °C) after 24 h and several cycles of thermocycling [16,32]. Like the previous subgroups, a statistical significance was also observed favouring the chlorhexidine group, with an increase in the microtensile bond strength by 6.71 MPa compared to the control group.

Thermocycling is one of the best known artificial aging methods. It helps to visualize the clinical behaviour of a material over time. In line with previous meta-analyses, chlorhexidine increased microtensile bond strength over time regardless of the artificial performance. Therefore, it supports the hypothesis that the effect of chlorhexidine on bonding strength is time-dependent [32].

Chlorhexidine can remain adhered to a substrate of demineralized and remineralized dentin irrespective of its concentration or application time [15,19]. This effectiveness may be due to its ability to inhibit the zinc- and calcium-dependent MMP mechanism [48]. Chlorhexidine competes for calcium in retention zones and bonds to negative phosphate or carboxyl groups [16] present in the mineralized dentin and in the collagen matrix. This prevents proteases or bacteria from forming bonds and reaching the collagen fibrils, which would explain how the hybrid layer remains intact after a lengthy period of time [53,54,55,56,57].

With the passage of time, the hybrid layer gets unstructured due to the activation and bonding of the MMPs in those areas of exposed dentine without infiltrating monomers [20]; however, if previously treated with chlorhexidine, the dentin remineralizes thanks to its cationic properties [49,55]. MMPs are only inhibited when chlorhexidine remains trapped in the dentinal matrix by 2-hydroxyethyl methacrylate (HEMA) [29,56,57]. When inhibited, the union of the MMPs to the non-infiltrated collagen fibers is avoided [15,49] increasing their action as time passes.

In 2014, a meta-analysis corroborated the linearity of chlorhexidine use and the increase in the stability of the hybrid layer in vitro studies, supporting the results obtained by the present study [45].

Also, the effect on the microtensile bonding strength of 2% chlorhexidine can be confirmed in this study, but the minimum concentration of chlorhexidine required to obtain positive results in the increase of bonding resistance is unknown. It can be affirmed that the association between the concentration of chlorhexidine and the bonding strength is not linear [44]. This controversy was highlighted in a review that found conflicting results in studies at different concentrations [58]. Therefore, further research should be carried out to clarify the minimum concentration needed to obtain favourable results in clinical practice.

To check whether the inhibitory effect of chlorhexidine was only related to time, the influence of adhesive type on the final microtensile bond strength was examined. The immediate and 24 h subgroups were studied because they had no statistical significance. This subgroup analysis was intended to analyze if there were any changes or statistical significance due to the brand of adhesive used.

All the adhesive products in this meta-analysis differed from each other (different commercial brands), but one characteristic they had in common was that the dentin had previously been prepared with orthophosphoric acid and washed with water following its application, since all these adhesives are etch-and-rinse systems with two or three steps.

After conducting the analysis based on the type of adhesive, the results were not statistically significant. It may therefore be stated that the adhesive did not influence the microtensile bond strength.

Although there are studies that confirm a difference in adhesion strength according to the adhesive used [32,40,42,44,45,47,48], all of them discuss the comparison between different systems of adhesives, such as the etch-and-rinse system with respect to the self-etch system. Adhesives by different commercial brands using the same system were never compared, which is the case for this investigation. This way, the possible effect of the adhesive on the final microtensile bond strength should be studied according to the brand of adhesive within the same system.

A positive effect of chlorhexidine on the prevention of hybrid layer degradation has also been observed in in vivo studies [59,60]. But all the studies have followed it up for a maximum period of 5 years in vitro [33]. Consequently, there is no certainty regarding the action of chlorhexidine on the interface after this time and whether the bond strength increases or decreases. This factor should be investigated further through studies conducted over longer periods.

Heintze [61] found a weak correlation of bond strength test results and clinical parameters in a systematic review. One should not rely solely on bond strength tests to predict the clinical performance of an adhesive system, but conduct other laboratory tests such as tests on the marginal adaptation of fillings in extracted teeth and the retention loss of restorations in non-retentive cavities after artificial aging.

The present review has shown high levels of heterogeneity, possibly because of the different sample sizes used in the included studies, the conditions affecting the teeth under study, their storage medium and time, and the presence of numerous variables. To control this heterogeneity, the studies were analyzed by follow-up times and the type of adhesive.

The measures taken to control publication bias were thorough searching in different sources (PubMed, Scopus, and Embase) with multiple search terms and a final manual search. Publication bias was assessed by a funnel plot, where some symmetry was observed by the classic fail-safe number, which returned a value of 1250 studies, and by Egger’s regression test, which gave *p* = 0.436, indicating that it had been controlled.

The Jadad scale for assessing the methodological quality of studies, like the PEDro scale, is designed for use with randomized clinical trials and does not possess the same accuracy for in vitro studies, such as those included in the present review. Blinding is not as relevant in in vitro studies as in randomized clinical trials, since the researchers’ judgement is unlikely to be distorted by subjective factors when working with specimens rather than with human beings.

In the same way, losses to follow-up do not have the same meaning when studying specimens or inanimate samples. Dental specimens do not suffer many losses through withdrawal. The only loss that can exceptionally occur, is through breakage, most of them occurring during the preparation of the sample before being evaluated by the test.

Due to the heterogeneity observed in the study, the level of evidence can be placed according to CEBM (Center for Evidence-Based Medicine) of Oxford at a level 2a, so it presents a moderate grade of recommendation. All the articles included in this review obtained medium–low quality scores, being a 3 the maximum score obtained by nine studies, as they only met the caution, because of the low accuracy of these scales for assessing in vitro studies. A quality scale validated for in vitro studies would be useful for future research.

## 5. Conclusions

Applying a 2% chlorhexidine solution after acid etching increased the microtensile bond strength. This effect was time-dependent, as it only proved significant in follow-up times of 6 months or more. The effect of chlorhexidine on the microtensile bond strength was not influenced by the type of adhesive. The results of the present study were based on in-vitro studies and they would not necessarily apply to in vivo conditions.

## Figures and Tables

**Figure 1 medicina-55-00769-f001:**
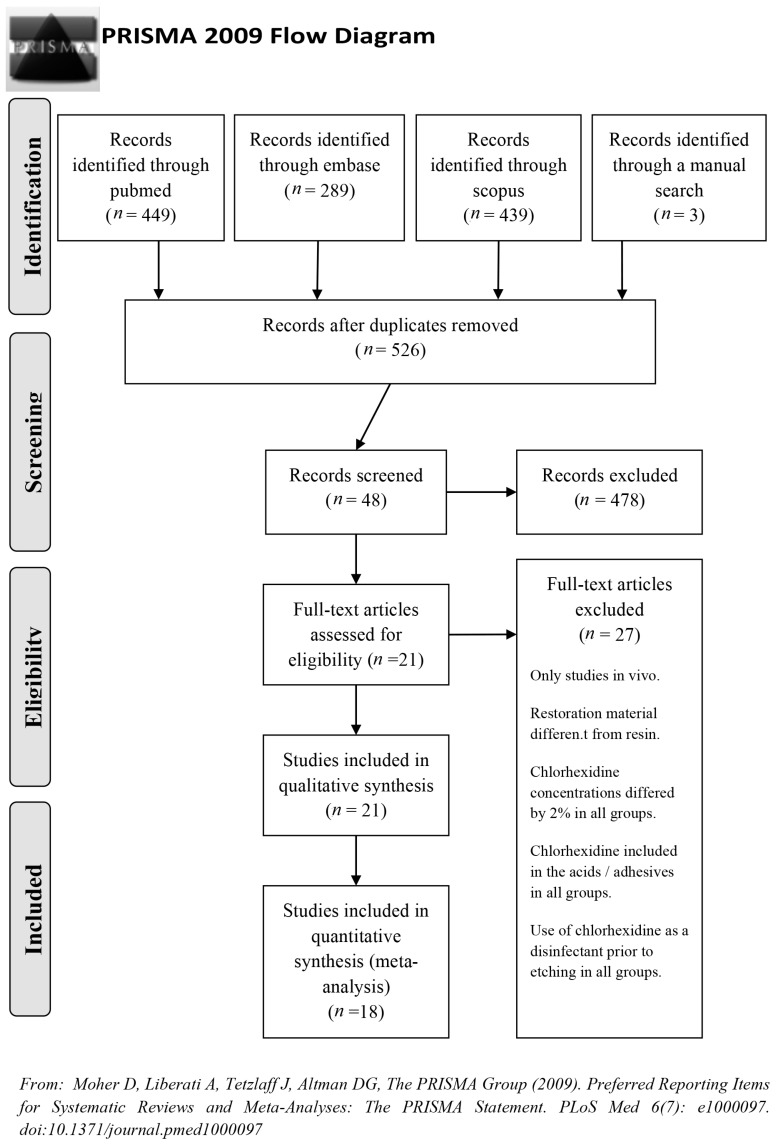
The PRISMA flow diagram. From: Moher D, Liberati A, Tetzlaff J, Altman DG, The PRISMA Group (2009). Preferred Reporting Items for Systematic Reviews and Meta-Analyses: The PRISMA Statement. [18].

**Figure 2 medicina-55-00769-f002:**
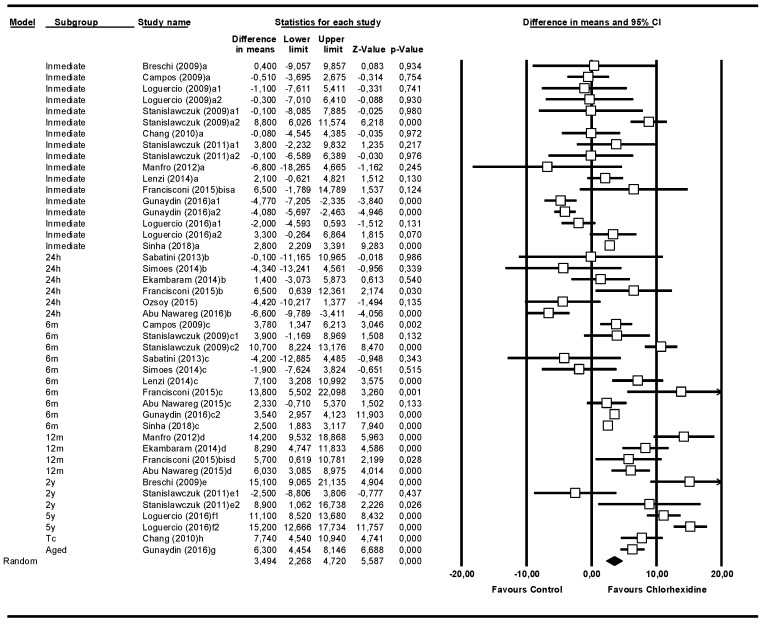
Overall forest plot of follow-up times: a - immediate; b - 24 h; c - 6 months; d - 12 months; e - 2 years; f- 5 years; g - aged, and h - thermally cycled. 1 and 2 indicates different types of adhesive within the same study.

**Figure 3 medicina-55-00769-f003:**
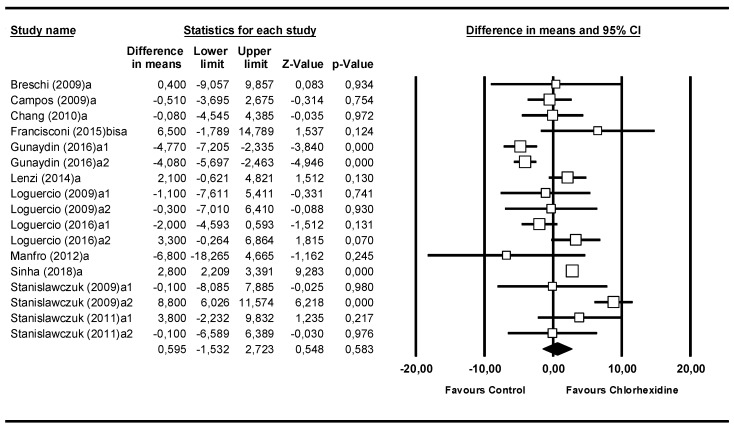
Forest plot by follow-up time, immediate.

**Figure 4 medicina-55-00769-f004:**
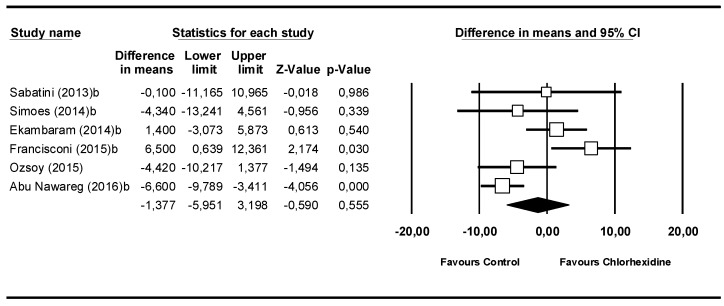
Forest plot by follow-up time, 24 h.

**Figure 5 medicina-55-00769-f005:**
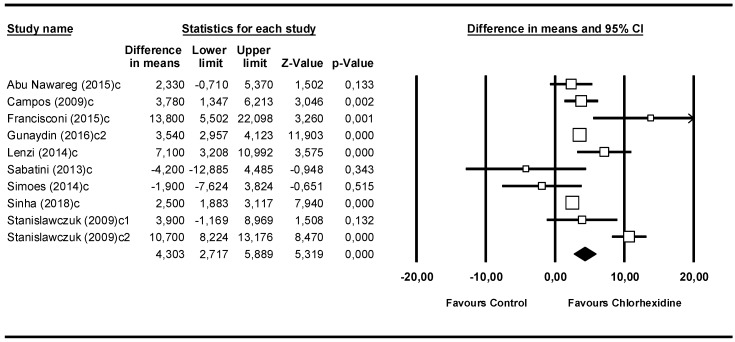
Forest plot by follow-up time, 6 months.

**Figure 6 medicina-55-00769-f006:**
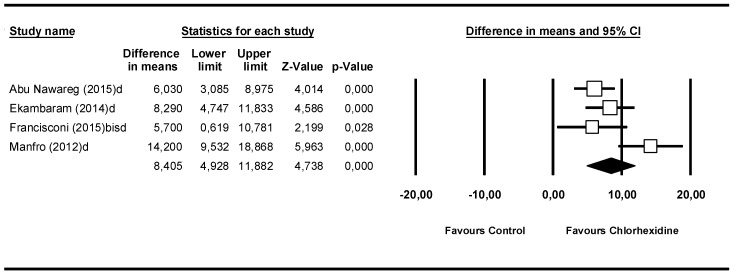
Forest plot by follow-up time, 12 months.

**Figure 7 medicina-55-00769-f007:**
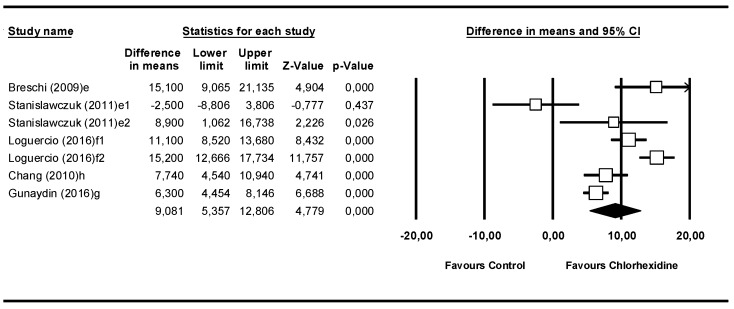
Forest plot by follow-up time, 2–5 years and aged or thermocycling.

**Figure 8 medicina-55-00769-f008:**
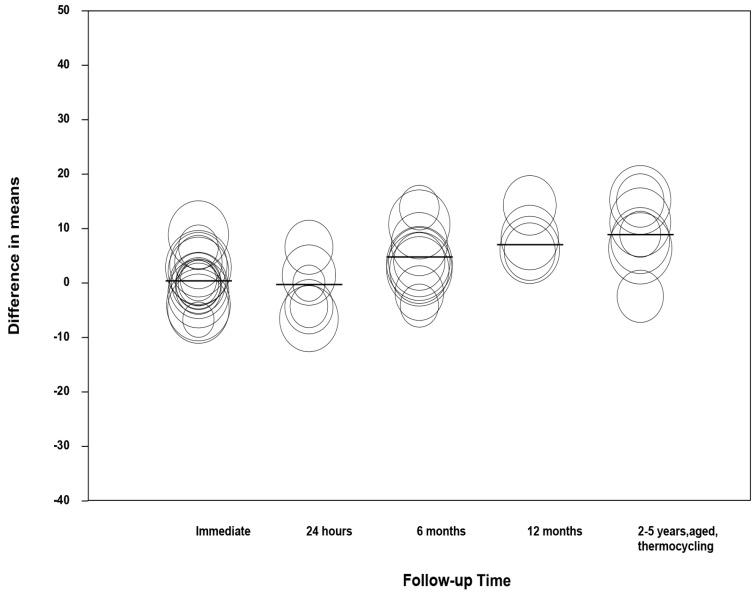
Scatter plot of difference in means on follow-up times.

**Figure 9 medicina-55-00769-f009:**
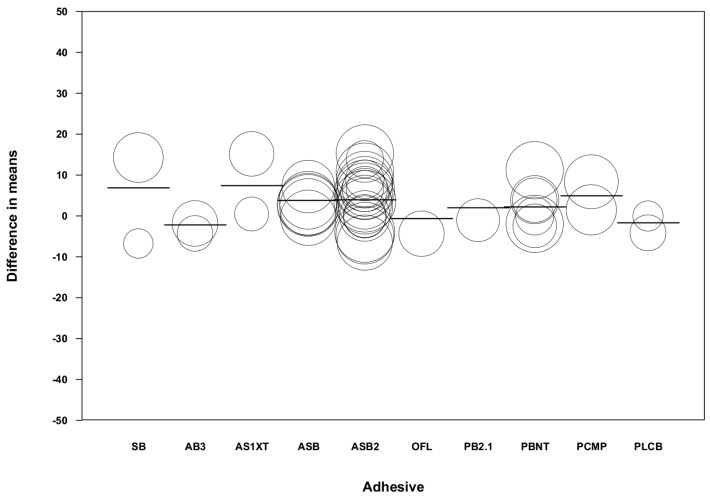
Scatter plot of difference in means on the type of adhesive.

**Figure 10 medicina-55-00769-f010:**
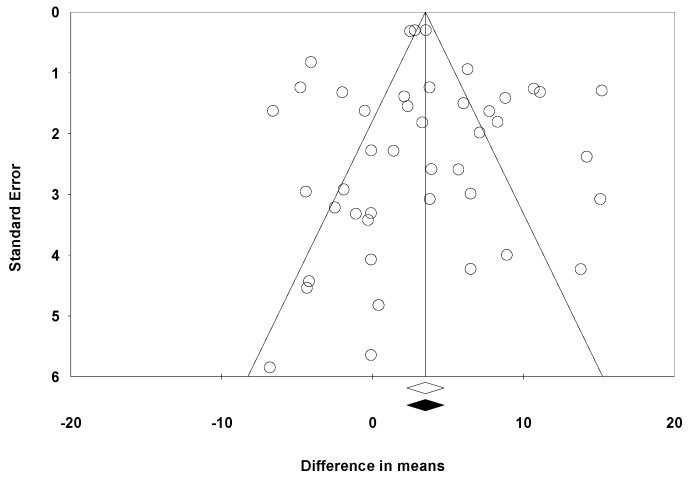
Funnel plot of the standard error by difference in the means.

**Table 1 medicina-55-00769-t001:** Studies included in the systematic review, detailed information about methodology, and results.

Author (yr)	Adhesive	Resin	N	Follow-Up Time	Control Group μTBS (Mpa) ± SD	2% Chlorhexidine Group μTBS (Mpa) ± SD
Carrilho (2007) [19]	SB	Filtek Z250	14	I; 6 m	data not available	data not available
Breschi (2009) [20]	AS1XT	Filtek Z250	8	I; 2 y	I: 40.8 ± 8.7	I: 41.2 ± 9.6
					2 y: 13.4 ± 4.9	2 y: 28.5 ± 7.2
Campos (2009) [21]	ASB	Filtek Z250	4	I; 6 m	I: 24.2 ± 1.65	I: 23.69 ± 2.80
					6 m: 13.65 ± 1.78	6 m: 17.43 ± 1.73
Komori (2009) [22]	SMP ASB 2	Filtek Z250	20	I; 6 m	data not available	data not available
Loguercio (2009) [14]	PB 2.1 ASB	Opallis	5	I; 6 m	PB 2.1_I: 32.4 ± 5.4	PB 2.1_I: 31.3 ± 5.1
					ASB_I: 41.5 ± 6.4	ASB_I: 41.2 ± 4.2
					PB 2.1_ 6 m: 21.2 ± 3.8	PB 2.1_6 m: 28.1 ± 4.5
					ASB_6 m: 25.4 ± 4.1	ASB_6 m: 37.6 ± 3.3
Stanislawczuk (2009) [15]	PB NT ASB 2	Opallis	14	I; 6 m	PB NT_I: 22.0 ± 9.7	PB NT_I: 21.9 ± 4.7
					ASB 2_I: 14.6 ± 3.1	ASB 2_I: 23.4 ± 2.1
					PB NT_6 m: 27.2 ± 6.1	PB NT_6 m: 31.1 ± 3.1
					ASB 2_6 m: 20.4 ± 2.1	ASB 2_6 m: 31.1 ± 2.6
Chang (2010) [16]	ASB 2	Filtek Z350	20	I; Tc	I: 29.4 ± 3.37	I: 29.4 ± 3.82
					Tc: 19.1 ± 2.49	Tc: 26.9 ± 2.6
Stanislawczuk (2011) [8]	PB NT ASB 2	Opallis	4	I; 2 y	PB NT_I: 29.0 ± 4.5	PB NT_I: 32.8 ± 4.2
					ASB 2_I: 32.3 ± 4.1	ASB 2_I: 32.2 ± 5.2
					PB NT_2 y: 29.0 ± 4.5	PB NT_2 y: 26.5 ± 4.6
					ASB 2_2 y: 17.2 ± 5.9	ASB 2_2 y: 26.1 ± 5.4
Manfro (2012) [23]	SB	Filtek Z250	7	I; 12 m	I: 50.8 ± 12.8	I: 44.0 ± 8.7
					12 m: 20.4 ± 3.7	12 m: 34.6 ± 5.
Sabatini (2013) [24]	PLCB	Filtek Z100	40	24 h; 6 m	24 h: 40.6 ± 15.7	24 h: 40.5 ± 8.5
					6 m: 46.4 ± 7.4	6 m: 42.2 ± 11.9
Simoes (2014) [25]	AB3	Filtek Z100	9	24 h; 6 m	24 h: 28.8 ± 11.0	24 h: 24.4 ± 8.0
					6 m: 19.4 ± 5.7	6 m: 17.5 ± 6.6
Ekambaram (2014) [26]	PCMP	Filtek Z250	12	24 h; 12 m	24 h: 25.2 ± 4.1	24 h: 26.6 ± 3.8
					12 m: 18.3 ± 4.0	12 m: 26.6 ± 1.9
Lenzi (2014) [27]	ASB	Filtek Z250	10	I; 6 m	I: 30.7 ± 2.2	I: 32.8 ± 3.8
					6 m: 24.2 ± 3.6	6 m: 31.3 ± 2.6
Francisconi (2015) [28]	ASB 2	Filtek Z350	14	24 h; 6 m	24 h: 34.7 ± 9.8	24 h: 41.2 ± 5.4
					6 m: 20.8 ± 14.4	6 m: 34.6 ± 6.6
Francisconi (2015)bis [29]	ASB 2	Filtek Z350	14	I; 6 m; 12 m	I: 34.7 ± 9.8	I: 41.2 ± 5.4
					6 m: 20.8 ± 14.4	6 m: 34.6 ± 6.6
					12 m: 7.6 ± 4.9	12 m: 13.3 ± 4.8
Ozsoy (2015) [30]	OFL	Tetric Ceram	6	24 h	24 h: 34.6 ± 4.5	24 h: 30.2 ± 2.4
Abu Nawareg (2016) [31]	ASB 2	Filtek Z350	12	24 h; 6 m; 12 m	24 h: 44.7 ± 4.0	24 h: 38.1 ± 3.9
					6 m: 34.5 ± 3.3	6 m: 36.8 ± 4.3
					12 m: 29.8 ± 3.5	12 m: 35.9 ± 3.9
Gunaydin (2016) [32]	ASB 2	Filtek Z250	20	I; Ag; I ivv; 6 m ivv	I: 36.1 ± 2.5	I: 31.4 ± 1.3
					Ag: 17.6 ± 1.7	Ag: 23.9 ± 1.23
					I ivv: 28.5 ± 2.3	I ivv: 24.4 ± 1.3
					6 m ivv: 16.3 ± 0.7	6 m ivv: 19.9 ± 0.7
Loguercio (2016) [33]	PB NT ASB 2	Opallis	28	I; 5 y	PB NT_I: 35.1 ± 3.1	PB NT_I: 33.1 ± 2.8
					ASB 2_I: 40.2 ± 3.3	ASB 2_I: 43.5 ± 3.5
					PB NT_5 y: 11.0 ± 2.7	PB NT_5 y: 22.1 ± 2.3
					ASB 2_5 y: 16.1 ± 2.1	ASB 2_5 y: 31.3 ± 2.7
Shadman (2018) [34]	SMP ASB	Filtek Z350	12	I; Tc	SMP_I: 13.9(11.4–16.4)	SMP_I: 13.7(10.0–17.3)
					ASB_I: 12.7(15.3–9.9)	ASB_I: 13.6(12.5–19.9)
					SMP_Tc: 14.6(12.4–16.7)	SMP_Tc: 11.5(8.5–15.7)
					ASB_Tc: 14.9(12.1–17.8)	ASB_Tc: 13.2(11.7–18.6)
Sinha (2018) [35]	ASB	Filtek Z350	40	I; 6 m	ASB_I: 16.8 ± 0.8	ASB_I: 19.6 ± 0.5
					ASB_6 m: 13.9 ± 0.8	ASB_6 m: 16.4 ± 0.6

Ag—aged; h—hours; I—immediate; ivv—in vivo; y—years; m—months; SB—Single Bond; ASB—Adper Single Bond; ASB 2—Adper Single Bond 2; SMP—Adper Scotchbond Multi-Purpose; PB NT—Prime & Bond NT; PB 2.1—Prime & Bond 2.1; AS1XT—Adper Scotchbond 1 XT; PLCB—Peak LC Bond; AB3—All Bond 3; PCMP—primer co-monomer pure; OFL—Optibond FL; Tc—thermocycling.

**Table 2 medicina-55-00769-t002:** Jadad quality scale.

Author (yr)	Randomization	Double-Blind	Withdrawals and Dropouts	Appropriate Randomization	Appropriate Blinding	Total
Carrilho (2007) [19]	1	0	0	1	0	2
Breschi (2009) [20]	1	0	1	1	0	3
Campos (2009) [21]	1	0	0	1	0	2
Komori (2009) [22]	0	0	1	0	0	1
Loguercio (2009) [14]	1	0	1	1	0	3
Stanislawczuk (2009) [15]	1	0	1	1	0	3
Chang (2010) [16]	1	0	0	1	0	2
Stanislawczuk (2011) [8]	1	0	1	1	0	3
Manfro (2012) [23]	1	0	0	1	0	2
Sabatini (2013) [24]	1	0	0	1	0	2
Simoes (2014) [25]	1	0	0	1	0	2
Ekambaram (2014) [26]	1	0	0	1	0	2
Lenzi (2014) [27]	1	0	0	1	0	2
Francisconi (2015) [28]	0	0	0	0	0	0
Francisconi (2015)bis [29]	1	0	1	1	0	3
Ozsoy (2015) [30]	0	0	1	0	0	1
Abu Nawareg (2016) [31]	1	0	1	1	0	3
Gunaydin (2016) [32]	1	0	0	1	0	2
Loguercio (2016) [33]	1	0	1	1	0	3
Shadman (2018) [34]	1	0	1	1	0	3
Sinha (2018) [35]	1	0	1	1	0	3

**Table 3 medicina-55-00769-t003:** Meta-regression with the difference of means as the dependent variable and the follow-up time and type of adhesive as covariates.

Covariate	Coefficient	95% Lower	95% Upper	Z-value	2-Sided *p*-value
Intercept	3.76	−3.89	11.41	0.96	0.335
Time: 24 h	−0.67	−5.65	4.32	−0.26	0.793
Time: 6 months	4.39	1.20	7.58	2.70	0.007 *
Time: 12 months	6.66	1.64	11.68	2.60	0.009 *
Time: 2–5 years and aged	8.50	4.89	12.11	4.61	<0.001 *
Adhesive: AB3	−9.06	−19.32	1.20	−1.73	0.084
Adhesive: AS1XT	0.54	−9.84	10.93	0.10	0.919
Adhesive: ASB	−3.05	−11.18	5.07	−0.74	0.462
Adhesive: ASB2	−2.94	−10.48	4.60	−0.76	0.445
Adhesive: OFL	−7.52	−19.53	4.49	−1.23	0.220
Adhesive: PB 2.1	−4.86	−16.70	6.98	−0.80	0.421
Adhesive: PBNT	−4.65	−12.94	3.63	−1.10	0.271
Adhesive: PCMP	−1.93	−10.73	6.87	−0.43	0.667
Adhesive: PLCB	−8.55	−19.85	2.75	−1.48	0.138

* Statistically significant.

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
