# Peer review of "Effect of 2% Chlorhexidine Following Acid Etching on Microtensile Bond Strength of Resin Restorations: A Meta-Analysis"

_1010-660X, 2019, doi:10.3390/medicina55120769_

Round 1

Reviewer 1 Report

General:

The micro tensile bond strength test has been the most popular method for evaluating the bond strength of adhesives across most bond testing laboratories. Therefore, a meta-analysis on this topic reporting on the effects of 2% chlorhexidine on long-term stability of bond strength seemed alluring. However, the manuscript is difficult to get through. In many cases, the citations do not match the interpretations made based on their results. Two of the references are in Spanish (1 and 3). Besides, the text needs to be checked for grammar corrections.   

Specific:

Abstract:

Line 21-22 grammatical error.

Introduction:

Two articles are cited as reference number 1. The first one is not in English.

Line 33-34: needs rephrasing. Reference number three is not in English.

Line 35: the role of chlorhexidine as a synthetic protease inhibitor was evaluated in the study by Carrilho et al., 2007 (Reference number 4), which is contrary to chlorhexidine’s disinfection role as mentioned in the text.

Line 48-50: needs rephrasing. The famous work of Nakabayashi et al., 1982 (reference number 11) had nothing to do with MMPs.

Line 51-53: needs rephrasing.

Line 54-61: needs rephrasing.

Line 64-65: The duration of chlorhexidine application (15 s) mentioned by the authors does not match with the citations mentioned (references 14 and 15), where chlorhexidine was applied for 60 s and 20 s respectively.

Line 66-68: needs rephrasing and grammar correction.

Line 70-71: reference number 10 should be cited as Pashley et al., which observed the effects of chlorhexidine for 250 days, not for 14 months as is mentioned in the text.

Line 71-73: The study (Gendron et al., 1999) cited as reference number 5 is an in vitro study, not a clinical study. If clinical suitability needed to be mentioned it’s better to use the source study.

Line 76-77: needs rephrasing. Chlorhexidine does not increase bond strength, rather known for providing long-term stability only.

Line 78-81: needs grammar correction.

Line 82-83: PICO needs to be rephrased as “do resin restorations pretreated with 2% chlorhexidine after etching have greater long-term micro tensile bond strength stability compared to their untreated counterparts?

As the micro tensile bond strength test is one of the centers of focus in this study a paragraph needs to be dedicated mentioning its inception and suitability over other bond strength testing methods in the introduction.  

Experimental section:

The organization of this study seems satisfactory.

Results:  

Well presented.

Discussion:

The interpretation of the citations needs to be checked.   

Conclusions:

Line 435-436: needs rephrasing.

References:

The number of references should be increased by including the inception and suitability of the micro tensile bond strength test over other bond strength testing procedures.

Author Response

RESPONSE TO REVIEWER 1 COMMENTS

General:

Point 1: The micro tensile bond strength test has been the most popular method for evaluating the bond strength of adhesives across most bond testing laboratories. Therefore, a meta-analysis on this topic reporting on the effects of 2% chlorhexidine on long-term stability of bond strength seemed alluring. However, the manuscript is difficult to get through. In many cases, the citations do not match the interpretations made based on their results. Two of the references are in Spanish (1 and 3). Besides, the text needs to be checked for grammar corrections.   

Response 1: Thank you for this comment. We have revised the manuscript and rewritten some parts so it is easier to understand. The references in Spanish have been eliminated. Grammar has been revised.

Specific:

Abstract:

Point 2: Line 21-22 grammatical error.

Response 2: Thank you for pointing this out. The error has been corrected.

Introduction:

Point 3: Two articles are cited as reference number 1. The first one is not in English.

Response 3: Thank you for the observation. This point has been corrected. References in Spanish (1 and 3) have been eliminated and new references have been added.

Point 4:  Line 33-34: needs rephrasing.

Response 4: Thank you for this comment, these lines have been rewritten: “The risk of resistance of oral bacteria to chlorhexidine and potential mechanisms conferring this resistance or even cross-resistances to antibiotics is little known. Moreover, there is also little awareness about the risk of chlorhexidine resistance among the dental community even though chlorhexidine has been widely used in dental practice as the gold-standard antiseptic [3-4].”

Point 5:  Reference number three is not in English.

Response 5: Thank you for this comment. Reference 3 has been eliminated and changed for a different one: Cieplik, F.; Jakubovics, N.S.; Buchalla, W.; Maisch, T.; Hellwig, E.; Al-Ahmad, A. Resistance toward chlorhexidine in oral bacteria- Is there cause for concern? Front microbiol 2019, 10, 587. DOI: 10.3389/fmicrob.2019.00587

Point 6: Line 35: the role of chlorhexidine as a synthetic protease inhibitor was evaluated in the study by Carrilho et al., 2007 (Reference number 4), which is contrary to chlorhexidine’s disinfection role as mentioned in the text.

Response 6: Thank you for this comment. This paragraph has now been changed so it is not as confusing.

Point 7: Line 48-50: needs rephrasing. The famous work of Nakabayashi et al., 1982 (reference number 11) had nothing to do with MMPs.

Response 7: Thank you for the suggestion. These lines have been rephrased.

Point 8: Line 51-53: needs rephrasing.

Response 8: Thank you for the suggestion. These lines have been rephrased.

Point 9: Line 54-61: needs rephrasing.

Response 9: Thank you for the suggestion. These lines have been rephrased.

Point 10: Line 64-65: The duration of chlorhexidine application (15 s) mentioned by the authors does not match with the citations mentioned (references 14 and 15), where chlorhexidine was applied for 60 s and 20 s respectively.

Response 10: Thank you for pointing this out. This error has been corrected: Reference 14 (15 or 60 seconds) and reference 15 (60 seconds).

Point 11: Line 66-68: needs rephrasing and grammar correction.

Response 11: Thank you for the suggestion. These lines have been rephrased and grammar has been revised.

Point 12: Line 70-71: reference number 10 should be cited as Pashley et al., which observed the effects of chlorhexidine for 250 days, not for 14 months as is mentioned in the text.

Response 12: Thank you for your comment. We have checked the references and the correct one is Carrilho (5) who conducted an in vivo study during 14 months. The error has been corrected.

Point 13: Line 71-73: The study (Gendron et al., 1999) cited as reference number 5 is an in vitro study, not a clinical study. If clinical suitability needed to be mentioned it’s better to use the source study.

Response 13: Thank you for pointing this out, the error has been corrected. The right reference is Carrilho (5).

Point 14: Line 76-77: needs rephrasing. Chlorhexidine does not increase bond strength, rather known for providing long-term stability only.

Response 14: Thank you for this comment. This phrase has been eliminated because it was repetitive and redundant.

Point 15: Line 78-81: needs grammar correction.

Response 15: Thank you for this comment. Paragraph has been rewritten.

Point 16: Line 82-83: PICO needs to be rephrased as “do resin restorations pretreated with 2% chlorhexidine after etching have greater long-term micro tensile bond strength stability compared to their untreated counterparts?

Response 16: Thank you for the comment. PICO has been rephrased as suggested.

Point 17: As the micro tensile bond strength test is one of the centers of focus in this study a paragraph needs to be dedicated mentioning its inception and suitability over other bond strength testing methods in the introduction.  

Response 17: Thank you for this suggestion. A paragraph and reference have been added.

Experimental section:

The organization of this study seems satisfactory.

Results:  

Well presented.

Discussion:

Point 18: The interpretation of the citations needs to be checked.

Response 18: Thank you for this comment. Discussion has been revised.

Conclusions:

Point 19: Line 435-436: needs rephrasing.

Response 19: This statement has been rephrased

References:

Point 20: The number of references should be increased by including the inception and suitability of the micro tensile bond strength test over other bond strength testing procedures.

Response 20: Thank you for this suggestion. Two new references on microtensile bond strength test have been added

Reviewer 2 Report

This is a very well conducted systematic review and meta-analysis regarding the effect of 2% chlorhexidine on the micro-tensile bond strength of etch-and-rinse adhesive systems. The method is well performed, the Discussion is sufficient and the conclusions are very interesting. I do not have any suggestions for improving this paper. Well done.

Author Response

Thank you for your comments. No changes to the manuscript were requested.